# Molecular Mechanisms of Chlorophyll Deficiency in *Ilex* × *attenuata* ‘Sunny Foster’ Mutant

**DOI:** 10.3390/plants13101284

**Published:** 2024-05-07

**Authors:** Yiping Zou, Yajian Huang, Donglin Zhang, Hong Chen, Youwang Liang, Mingzhuo Hao, Yunlong Yin

**Affiliations:** 1College of Forestry, Nanjing Forestry University, Nanjing 210037, China; yiping200889@126.com (Y.Z.);; 2Institute of Botany, Jiangsu Province and Chinese Academy of Sciences (Nanjing Botanical Garden Mem. Sun Yat-Sen), Nanjing 210014, China; 3Jiangsu Qinghao Landscape Horticulture Co., Ltd., Nanjing 211225, China; 4Department of Horticulture, University of Georgia, Athens, GA 30602, USA

**Keywords:** chlorophyll deficient, chlorophyll synthesis, chloroplast development, carotenoid biosynthesis, photosynthesis, transcriptome

## Abstract

*Ilex* × *attenuata* ‘Sunny Foster’ represents a yellow leaf mutant originating from *I.* × *attenuata* ‘Foster#2’, a popular ornamental woody cultivar. However, the molecular mechanisms underlying this leaf color mutation remain unclear. Using a comprehensive approach encompassing cytological, physiological, and transcriptomic methodologies, notable distinctions were discerned between the mutant specimen and its wild type. The mutant phenotype displayed aberrant chloroplast morphology, diminished chlorophyll content, heightened carotenoid/chlorophyll ratios, and a decelerated rate of plant development. Transcriptome analysis identified differentially expressed genes (DEGs) related to chlorophyll metabolism, carotenoid biosynthesis and photosynthesis. The up-regulation of *CHLD* and *CHLI* subunits leads to decreased magnesium chelatase activity, while the up-regulation of *COX10* increases heme biosynthesis—both impair chlorophyll synthesis. Conversely, the down-regulation of *HEMD* hindered chlorophyll synthesis, and the up-regulation of *SGR* enhanced chlorophyll degradation, resulting in reduced chlorophyll content. Additionally, genes linked to carotenoid biosynthesis, flavonoid metabolism, and photosynthesis were significantly down-regulated. We also identified 311 putative differentially expressed transcription factors, including *bHLH*s and *GLK*s. These findings shed light on the molecular mechanisms underlying leaf color mutation in *I.* × *attenuata* ‘Sunny Foster’ and provide a substantial gene reservoir for enhancing leaf color through breeding techniques.

## 1. Introduction

Leaf hue is a significant trait in ornamental plants, drawing public interest and scholarly inquiry [1]. Leaf color mutants present a prime opportunity to delve into chlorophyll (Chl) metabolism and photosynthesis while unraveling the intricacies of chloroplast structure, development, and functionality [2,3]. The diverse leaf pigments, notably Chl, carotenoids (Car), and anthocyanins, contribute to the vibrant hues observed in plant foliage [4,5]. Chl, as the primary pigment responsible for leaf greenness, holds particular importance, with leaf color mutants often exhibiting Chl deficiency [6]. These mutations significantly affect pigment synthesis and chloroplast development, leading to alterations in pigment quantity, ratio, and leaf color phenotype [7,8]. Albino rice leaves exhibit markedly reduced Chl concentrations compared to their green counterparts [9]. Correspondingly, the yellow-green mutant *Cmygp* in melon manifested a Chl content of approximately 57.14% in comparison to the green phenotype [10]. Conversely, the deep green mutant *ZD131* in cotton demonstrates a substantially elevated Chl content, reaching approximately 115% of its wild type [11].

The biosynthesis of Chl entailed a sophisticated process involving over twenty enzymatic steps. It commenced with the conversion of glutamate to 5-aminolevulinic acid (ALA). Subsequently, ALA underwent a series of conversions, including porphobilinogen (PBG), uroporphyrinogen III (Urogen III), coproporphyrinogen III (Coprogen III), protoporphyrin IX (Proto IX), magnesium-protoporphyrin IX (Mg-Proto IX), protochlorophyllide (Pchlide), ultimately culminating in the synthesis of chlorophyll b and chlorophyll a [12]. Due to the intricate enzymatic processes governing Chl biosynthesis, disruptions at any stage led to a deficiency of green pigmentation in plant leaves and various leaf color phenotypes, as evidenced by mutants like *ygl1* in *Brassica napus* and *YX-yl* in *Panicum miliaceum* [13,14]. Analysis of these Chl precursors is crucial for detecting metabolic abnormalities in Chl biosynthesis [15]. When a process in Chl synthesis is inhibited, precursor substances upstream of the blocked site tend to accumulate significantly, whereas those downstream of the obstruction exhibited a marked decrease [16]. For example, the yellow mutant *lrysl1* identified in *Lilium regale* impeded the synthesis of ALA, consequently leading to Chl deficiency [17]. Similarly, disruptions in Chl synthesis occurred during the conversion of Proto IX to magnesium protoporphyrin in the rice yellowish-green leaf mutants *Chlorina-1* and *Chlorina-9* [18]. Multiple genes, including *HEMC*, *CHLH*, *HEMD*, *CAO* and *SGR*, are directly linked to Chl synthesis, with mutations in these genes resulting in altered Chl levels [19,20]. A point mutation in *CsCHLI*, a subunit of magnesium chelatase I crucial for catalyzing the insertion of Mg^2+^ into protoporphyrin IX within the Chl synthesis pathway, led to the emergence of a golden cucumber leaf mutant [21].

Moreover, Chl-deficiency mutants often show distinct variations in thylakoid structure, hampering photosynthesis due to abnormal chloroplast and membrane structures [22,23]. Yellow-green leaves and malformed chloroplasts with limited membranes, incomplete stacking, and decreased photosynthesis are observed in the *Petunia* mutant E5059, characterized by Chl deficiency [24]. Similarly, in the *Arabidopsis var2* mutant, regions displaying green sectors with typical chloroplasts coexisting alongside variegated sectors exhibited abnormal chloroplasts lacking internal membrane structures and hampered photosynthesis [25].

In recent years, researchers have employed various innovative techniques, such as transcriptomics, proteomics, metabolomics, and integrated approaches, to unravel the complex mechanisms behind leaf color mutation [26,27]. For instance, physiological and transcriptome analysis of *I.* × ‘Whoa Nellie’ unveiled alterations in chloroplast ultrastructure and reduced gene expression linked to Chl biosynthesis and photosynthesis, leading to Chl deficiency in the mutant leaves [24]. Similarly, transcriptomic analysis and weighted gene co-expression network analysis (WGCNA) profiling in *Ginkgo biloba* identified differentially expressed genes (DEGs) involved in pigment metabolism and putative transcription factors (TFs) associated with its leaf coloration [28]. Through an integrated transcriptomic and metabolomic investigation in *Cucumis melo*, enzyme genes and metabolites predominantly associated with central carbon metabolism, photosynthesis, flavonoid metabolism, and energy metabolism were identified [29]. Investigations utilizing iTRAQ-based quantitative proteomics analysis identified 168 differentially accumulated proteins that are involved in photosynthesis, Chl, and Car metabolism in the leaf color formation of the hybrid paper mulberry [30]. These studies collectively highlighted the intricate nature of leaf color mutation mechanisms observed from various plant species.

Leaf color mutations have been intensively studied across a range of plant species like tea [31], rice [8], and wheat [32]. The Chl synthesis of albino ‘Huangjinya’ tea and the formation of photosynthesis-related proteins were inhibited, resulting in photosynthetic efficiency decrease [33]. Similarly, the virescent yellow-leaf mutant in rice exhibited compromised chloroplast development and decreased Chl levels [34]. The albino cucumber cultivar ‘g32’ obtained an albino phenotype that contributed to the disruption of chloroplast development [23]. Due to impaired photosynthesis, the yellow *NAU31* mutant had impaired chloroplast structure and reduced Chl and Car content [35]. Although, in recent years, ornamental plants have also witnessed the emergence of Chl-deficiency mutants, including species like *G. biloba* [36], *Quercus shumardii* [37], *Ulmus pumila* [38], *Lagerstroemia indica* [39], and *Acer palmatum* [40]. However, there is a paucity of comprehensive investigations into the molecular mechanisms underlying ornamental leaf color mutants. Among these, hollies (genus *Ilex*) stand out for their highly valued traits in landscape design, including diverse fruit colors, leaf textures and colors [41]. The vibrant color of holly leaves is often regarded as a defining characteristic by many gardeners [42]. Among the extensive range of *Ilex* cultivars, more than 165 are colored-leaf varieties [43,44]. Nevertheless, research on the leaf mutation mechanism in hollies is scarce.

In our study, we delve into an important yellow leaf holly variety known as *I*. × *attenuata* ‘Sunny Foster’, a natural bud mutant of the wild type *I*. × *attenuata* ‘Foster #2’. It is a well-known Foster Holly for its attractive accent and bright yellow flush in direct sunlight, and received the Holly Society of America “Gene Eisenbeiss Holly of the Year” award in 2006 [45]. As an excellent ornamental plant, it has been popularized and planted in most areas of America for its color leaves, fruits, tree form, good vigour, cold hardiness, and easy culture [46,47]. Although the cytological, physiological, and transcriptomic properties of this mutant remain unexplored, our investigation sought to quantify the disparities in these aspects between the mutant and its wild-type counterpart. By employing a multidisciplinary approach encompassing advanced microscopy techniques, physiological measurements, and high-throughput RNA sequencing, we aimed to provide a thorough comprehension of the determinants influencing leaf coloration. The findings greatly enhanced our comprehension of the fundamental mechanisms governing leaf color mutation in ornamental plants and furnished valuable resources to inform and guide ornamental plant breeding efforts.

## 2. Results

### 2.1. Yellow Hue and Brighter Color in the Mutant

Leaf color differences were quantitatively assessed using parameters including L*, a*, b*, a*/b*, H° and C. Yellow leaves exhibited significantly higher L* and C values compared to green leaves, indicating a brighter appearance. Additionally, both green and yellow leaves had positive b* values, situating them in the red-yellow region of the CIELab color map (Figure 1C). Notably, yellow leaves displayed a lower H° value (87.32°) compared to green leaves (118.40°), suggesting a shift towards a yellow hue in the mutant. The negative a*/b* ratio in green leaves signified their green coloration, with a larger absolute value indicating a deeper green tone (Figure 1A,B).

### 2.2. Reduced Chl Synthesis in the Mutant

In yellow leaves, there were significant reductions in SPAD value, Chl a, Chl b, Chl a + b, Car, and flavonoid contents compared to green leaves, showing decreases of 97.63%, 98.53%, 95.28%, 87.73%, and 97.81%, respectively (Figure 1D,F,G). However, no significant difference was observed in anthocyanin content between the two varieties (Figure 1F). Notably, the ratios of Car/Chl a, Car/Chl b, and Car/Chl a + b exhibited a significant increase in yellow leaves, approximately 10.48, 3.24, and 6.98 times the levels observed in green leaves (Figure 1E).

The contents of all seven Chl precursors were diminished in yellow leaves in contrast to green leaves. While the differences in the levels of ALA and PBG between yellow and green leaves were not significant, the disparities in the levels of Urogen III, Coprogen III, Proto IX, Mg-Proto IX, and Pchlide were statistically significant (Figure 1H), mirroring the trends observed in Chl contents (Figure 1F).

### 2.3. Varied Leaf Anatomical Structure and Impaired Chloroplast Ultrastructure in the Mutant

The epidermal micromorphological characteristics observed under scanning electron microscopy were comparable between the mutant and wild-type leaves. The upper epidermal cells in both leaf types exhibited inconspicuous margins, accompanied by fine lines or distinct ridges on the cuticle surface. Additionally, no epidermal hairs were observed (Figure 2A,D). The stomatal apparatus was exclusively distributed in the lower epidermis and displayed an irregular shape (Figure 2B,E). The stomata were oval-shaped and composed of two guard cells and two accessory guard cells (Figure 2C,F). Moreover, the inner margin of the outer arch cover of the stomata appeared to be nearly smooth (Figure 2C,F). While the distribution pattern and form of stomata in yellow leaves were similar to those in green leaves, significant micromorphological changes were observed between the two leaf types. Although the stomatal length and width did not show significant differences between *I.* × *attenuata* ‘Foster #2’ (WT) and *I.* × *attenuata* ‘Sunny Foster’ (YT), yellow leaves exhibited a markedly lower stomatal density compared to green leaves (Appendix A). Additionally, while the stomata of green leaves were mostly open (Figure 2E,F, Appendix A), the majority of stomata in yellow leaves were almost fully closed (Figure 2B,C, Appendix A). These observations implied a potential impairment in heat dissipation through the stomatal regulation in the YT mutant.

Both types of leaves featured an apparent upper epidermis, palisade tissue, spongy tissue, and lower epidermis. In the wild-type green leaves, the majority of chloroplasts were situated within mesophyll cells, comprising both palisade mesophyll cells and spongy mesophyll cells (Figure 3E). Notably, the palisade tissue housed a higher concentration of chloroplasts than the spongy tissue. The internal structure of yellow leaves closely resembled green leaves (Figure 3A). However, in yellow leaves, the sponge tissue appeared more spacious compared to wild-type plants. The relatively lighter color of cross-sections from yellow leaves was attributed, at least in part, to the reduced size of chloroplasts within these leaves. Additionally, a diminished Chl content also contributed to the lighter coloration.

There were notable distinctions in chloroplast ultrastructure between the green and yellow leaves (Figure 3B–D,F–H, Table 1). Chloroplasts in WT plants displayed spindle or spindle-shaped morphology, characterized by a well-organized inner membrane system with regularly distributed granal thylakoids containing abundant granal lamellae, starch granules, and limited plastoglobuli presence (Figure 3F–H). Green leaves typically contained an average of approximately 9.6 chloroplasts per cell (Table 1). On the contrary, chloroplasts in yellow leaves displayed swelling, lacked distinct grana, had unclear or absent stromal lamellae, and were densely packed with numerous vesicles and plastoglobuli (Figure 3B–D, Table 1). Yellow leaves exhibited an average of approximately 3.9 chloroplasts per cell (Table 1), indicating a reduction in number and altered morphology. Analysis from Table 1 revealed that chloroplasts in green leaves had a length of approximately 5.98 μm and a width of about 2.31 μm, resulting in a length/width ratio of about 2.67. Conversely, chloroplasts in yellow leaves had a length of approximately 6.46 μm and a width of about 5.34 μm, yielding a length/width ratio of about 1.25, indicating an increased chloroplast volume in yellow leaves. While green tissue contained approximately 2.1 starch granules per chloroplast, the absence of starch granules was notably observed in chloroplasts of yellow leaves (Table 1). In summary, alterations in the number, size, and morphology of chloroplasts in yellow leaves are evident, alongside incomplete development of chloroplasts.

### 2.4. Decreased Photosynthetic Capacity and Delayed Growth in the Mutant

Compared to the wild-type plants, the YT plants displayed significant reductions of 98.10% in net photosynthetic rate (P_n_), 86.26% in stomatal conductance (G_s_), and 79.90% in transpiration rate (E). Yellow leaves exhibited a 35.82% higher intercellular CO_2_ concentration (C_i_) compared to the WT (Table 2). The decrease in P_n_ and G_s_ led to a reduced capacity for CO_2_ absorption, thereby limiting photosynthesis. However, the C_i_ significantly increased in mutant plants (Table 2), indicating a decreased rate of CO_2_ utilization in mutants. The mutant exhibited reductions of 29.60% in fluorescence origin (Fo), 75.81% in fluorescence maximum (Fm), 94.25% in variable fluorescence (Fv), 76.39% in maximal quantum yield of PSII photochemistry (Fv/Fm), 77.08% in actual photosynthetic efficiency of PSII (ΦPSII), 95.83% in non-photochemical quenching (NPQ), and 88.24% in regulatory energy dissipation quantum yield (Y(NPQ)) compared to its wild type plants (Table 2). These reductions suggest that the mutant had a compromised self-protective capacity and reduced photochemical efficiency. Non-regulatory energy dissipation quantum yield (Y(NO)) in the mutant was 59.77% higher compared to that of WT, indicating increased photodamage in the mutant. In combination, these findings pointed towards a deficiency in the photosynthetic capacity of the yellow mutant plants.

Mutant 2-year-old plants exhibited significantly shorter heights compared to their wild-type counterparts, measuring only 75% of the wild-type plants (Figure 4A). The fresh mass of mutant plants’ leaves, stems, and roots decreased dramatically by 23.81%, 35.69%, and 48.21%, respectively (Figure 4B). The dry mass of mutant plant leaves, stems, and roots significantly decreased by 5.83%, 40.42%, and 52.22%, respectively (Figure 4C). The levels of soluble protein in mutant plant leaves significantly decreased by 4.21%, while the levels of soluble sugar significantly increased by 8.05% (Figure 4D,E). These findings suggest that mutant plant growth was significantly deterred.

### 2.5. Transcriptomic Modifications in Pigment Metabolism and Photosynthesis in the Mutant

After rigorous quality control, we obtained a total of 260,393,656 clean reads, with 127,328,916 reads derived from the WT and 133,064,740 reads from the YT mutant. These clean data exhibited Q20 and Q30 percentages surpassing 96.77% and 91.39% and had a GC content ranging from 44.08% to 44.52%, indicating high quality. The clean data exhibited mapping rates to the *I. latifolia* genome ranging from 67.44% to 68.32% (Appendix A). Replicate samples displayed high correlation, ensuring the uniformity of RNA-seq data (Figure 5A). Appendix A provides an overview of the RNA-Seq data.

YT and WT exhibited a total of 3024 DEGs, with 1212 genes showing up-regulation and 1812 genes displaying down-regulation (Figure 5B,C). For deeper functional understanding, we conducted Gene Ontology (GO) and Kyoto Encyclopedia of Genes and Genomes (KEGG) enrichment analyses. In the biological process category of the GO analysis, the most significantly enriched terms were “cellular carbohydrate metabolic process” (GO:0044262), “polysaccharide metabolic process” (GO:0005976), and “cellular glucan metabolic process” (GO:0006073). Moving to the cellular component category, terms like “Cell wall” (GO:0005618), “external encapsulating structure” (GO:0030312), and “apoplast” (GO:0048046) took prominence. Within the molecular function category, “Zinc ion binding” (GO:0008270), “glucosyltransferase activity” (GO:0046527), and “hydrolase activity, acting on glycosyl bonds” (GO:0016798) stood out prominently (Figure 5D,E). These results suggest alterations in carbohydrate metabolism, cell membrane composition, and catalytic activities in the YT mutant, possibly contributing to its phenotype.

The KEGG pathway enrichment analysis revealed that the DEGs were distributed across 122 pathways. Notably, pathways such as “starch and sucrose metabolism” (ko00500), “phenylpropanoid biosynthesis” (ko00940), “porphyrin metabolism” (ko00860), “biosynthesis of co-factors” (ko01240) and “biosynthesis of various plant secondary metabolites” (ko00999) showed significant enrichment (*Padj* < 0.05). Additionally, “flavonoid biosynthesis” (ko00941), “photosynthesis-antenna proteins” (ko00196) and “photosynthesis” (ko00195) were included in the top 20 pathways (Figure 5F). Given the decreased Chl, Car, flavonoid contents and impaired photosynthetic efficiency in the yellow mutant, the pathways associated with porphyrin metabolism, Car biosynthesis, flavonoid biosynthesis, and photosynthesis warranted particular attention. Specifically, we identified 18 DEGs involved in the “porphyrin metabolism” pathway, seven DEGs in “carotenoid biosynthesis”, 10 DEGs linked to “flavonoid biosynthesis”, 16 DEGs related to the “photosynthesis”, and seven DEGs associated with “photosynthesis-antenna proteins” (Appendix A).

### 2.6. Altered Chl Biosynthesis in the Mutant

Through KEGG enrichment analysis, we identified 18 DEGs encoding enzymes closely associated with porphyrin and Chl metabolism (Figure 6, Appendix A). Among these DEGs, 15 genes related to Chl synthesis (e.g., *GSA*, *HEMC*, *CHLD*, *PORA*) and 2 genes linked to Chl degradation (e.g., *SGR*) were up-regulated, while *HEMD* (encoding S-adenosyl-L-methionine-dependent uroporphyrinogen III methyltransferase) associated with Chl synthesis showed down-regulation. Notably, the concentrations of Chl precursors, spanning from ALA to PBG, exhibited no substantial disparities between WT and YT. However, Urogen III, Coprogen III, Proto IX, Mg-Proto IX, and Pchlide displayed significant decreases in YT, suggesting an impediment from PBG to Urogen III synthesis. These findings indicate that the inhibition of Chl synthesis in YT plants was attributed to the decreased expression of *HEMD*, while the increase in *SGR* expression likely contributed to the reduction in Chl levels. The increased expression of other genes involved in Chl synthesis served as a compensatory mechanism arising from the deficiency of Chl.

Gene Set Enrichment Analysis (GSEA) analysis revealed markedly enriched molecular pathways, including RNA polymerase (ath03020), ribosome (ath03010), homologous recombination (ath03440), porphyrin metabolism (ath00860), and pantothenate and CoA biosynthesis (ath00770) in YT (nominal *p*-value < 0.05, NES > 1.5, FDR *q*-value < 0.25) (Appendix A). Beyond the previously discussed up-regulated DEGs, the enriched porphyrin metabolism pathway encompassed 14 DEGs, including *CHLG*, *CHLP*, *CLH2*, *COX10*, *HEMF*, *HCAR*, *HY2*, *NOL*, *PORA*, *PPOX1* (Appendix A, Appendix A). This underscores the significance of Chl metabolism-related genes in influencing leaf coloration in YT. The pronounced enrichment *COX10* (encoding heme synthase) (ES = 0.618) highlights possible disruptions in the Chl-heme metabolic pathway, thus emphasizing their pivotal role in determining leaf coloration in *I.* × *attenuata* ‘Sunny Foster’.

### 2.7. Down-Regulated Carotenoid and Flavonoid Biosynthesis in the Mutant

Six DEGs encoding enzymes crucial for Car biosynthesis were identified, with four being down-regulated and two up-regulated in the mutant. Down-regulated DEGs included *CCS*, *NCED*, *and CYP707A1*, while *LCYE* and *CYP707A2* were significantly up-regulated in the mutant (Figure 7, Appendix A). The observed decrease in Car content in yellow leaves suggested an association with the down-regulation of genes involved in Car metabolism, indicating a disruption in Car biosynthesis pathways in the mutant plants.

Given YT’s significantly decreased flavonoid content compared with WT, we subsequently analyzed the flavonoid biosynthesis pathway. We pinpointed 10 DEGs linked to flavonoid biosynthesis, with all exhibiting down-regulation in the mutant, including *CYP92C6*, *CAMT*, *CYP98A2*, *HCT*, *ANR*, and *CHI* (Appendix A).

### 2.8. Down-Regulated DEGs in Photosynthesis and Chloroplast Development

The aberrant chloroplast ultrastructure and photosynthesis in the YT mutant prompted further investigation into the molecular mechanisms underlying these phenotypic changes. Significant differential expression was observed in 41 genes related to photosynthesis and chloroplast development when comparing the mutant with normal green leaves, as depicted in the heatmap presented in Figure 8 and Appendix A. Specifically, while two DEGs associated with the PSII reaction center subunit (*psbC*) demonstrated reduced expression, two DEGs involved in the PSI reaction center subunit (*psaA* and *psaB*) and three DEGs (*psbQ*, *psbK*, and *psbE*) exhibited higher expression levels in the mutant (Figure 8B,C, Appendix A). One cytochrome b6/f complex gene (*petD*), one photosynthetic electron transport gene (*petF*), and four F-type ATPase genes (*atpI*, *atpA*, *atpB*, *atpE*) exhibited higher expression levels in the mutant. On the contrary, the mutant exhibited reduced expression levels in three photosynthetic electron transport genes (*petF* and *petH*) (Figure 8D–F, Appendix A). Three genes involved in carbon fixation in photosynthesis had increased transcription, while six genes were significantly down-regulated in the YT mutant, affecting carbon assimilation. The downregulation of these genes weakened CO_2_ fixation (Figure 8G, Appendix A). All nine genes associated with nitrogen metabolism were down-regulated (Figure 8H, Appendix A). The down-regulation of these genes suggested a potential impairment of photosynthetic activity in the mutant plants.

Differential expression of the light-harvesting chlorophyll protein complex was also observed between YT and WT in the photosynthesis-antenna protein pathway. Specifically, four DEGs encoding light-harvesting complex II Chl a/b binding proteins (LHCB) and three DEGs encoding light-harvesting complex I Chl a/b binding proteins (LHCA) were significantly up-regulated in the mutant (Figure 8A, Appendix A). The increased expression of these key genes indicated a compensatory regulatory mechanism to efficiently capture and transfer light energy for photosynthesis in the mutant.

### 2.9. Differentially Expressed Transcription Factors in the Mutant

A comprehensive analysis revealed 311 DEGs encoding putative transcription factors (TFs), spanning 43 TF families (Figure 9, Appendix A). The analysis highlighted nine predominant TF families among these, namely *bHLH*s, *ERF*s, *NAC*s, *MYB-related*, *MYB*s, *G2-like*s (*GLK*), *C2H2*s, *FAR1*s, and *WRKY*s (Figure 9). The *bHLH* family was most abundant, comprising 29 TFs and representing 9.3% of the total. Out of these, 16 TFs displayed decreased expression, while 13 TFs demonstrated increased expression in the mutant. Additionally, after *bHLH*, the *ERF* (26 TFs, 8.4%), *NAC* (22 TFs, 7.1%), *MYB-related* (21 TFs, 6.8%), *MYB* (18 TFs, 5.8%), and *GLK* (17 TFs, 5.5%) families displayed the next highest representation (Figure 9, Appendix A). It is worth mentioning that 13 *GLK* TFs demonstrated decreased expression, encompassing *GLK1* and *GLK2*, whereas 4 TFs displayed increased expression in the mutant. These results suggest potential disruptions in chloroplast development, which could play a role in the manifestation of the observed yellow phenotype in the mutant.

### 2.10. qRT-PCR Validation of the Candidate DEGs

The DEGs identified through RNA-Seq analysis were validated using the qRT-PCR method. We selected a range of genes implicated in Chl metabolism, Car metabolism, PSI, PSII, and TFs for validation (Appendix A). A total of 12 DEGs were examined, and their expression patterns aligned with the transcriptome data (Appendix A, Appendix A). This consistency between the qRT-PCR results and transcriptome data affirmed the credibility of the transcriptome data and validated its use for this study.

## 3. Discussion

Mutants lacking Chl offer essential contributions to scientific research and practical applications, especially concerning woody plant mutants with altered leaf coloration [48,49]. Holly leaf color mutants have received significant interest owing to their distinct foliage hues and economic and practical utility [42]. In this study, we investigated a significant Chl-deficient holly cultivar exhibiting a yellow leaf phenotype. Through comprehensive cytological, physiological, and transcriptomic analyses, we aimed to elucidate its distinctive characteristics.

The differences in coloration between the mutant leaves and those of the wild type highlight the exceptional nature of this leaf color mutation, reflecting the diverse leaf coloration observed among the plants [50]. The significant reductions in Chls contents in yellow leaves signify a substantial impairment in Chl biosynthesis. Specifically, the decrease in Chl b exceeded that of Chl a, classifying this mutant as a Chl b-deficient type according to Falbel’s classification [51]. Similar observations have been reported in a yellow mutant of *Q. shumardii*, which exhibited higher L* and C values, lower Chl content, and a pronounced decline in Chl b compared to Chl a [37]. The significantly higher ratios of Car/Chls in yellow leaves could be the direct reason for the mutant’s yellow appearance, as seen in yellow mutants of *G. biloba* [15], *Brassica napus* [14], and *L. indica* [52].

The process of Chl biosynthesis is essential for both photosynthesis and plant development, given the critical role of Chl molecules. They play pivotal roles in absorbing light energy, facilitating electron transfer within the photosynthetic electron transport chain, and initiating reactions that separate charges within reaction centers [53,54]. Photosynthesis, in turn, is fundamental for synthesizing organic compounds essential for plant growth and development [55]. This process occurs on the thylakoid membranes of chloroplasts, which are structured into regularly stacked grana in higher plants, optimizing light harvesting and energy conversion efficiency [56]. Therefore, leaf color mutants often exhibit structural alterations in thylakoid organization, such as reduced grana stacks and irregular arrangement, indicating an underdeveloped chloroplast ultrastructure and potential loss of photosynthetic capacity [26,57]. For instance, in the *pgb* tea Chl-deficient mutant, mature leaves exhibited significantly fewer chloroplasts with defective morphology, lacking stacked thylakoid structures but containing numerous giant plastoglobuli, along with reduced osmiophilic granules and absence of starch granules in the plastids [58]. Similarly, in this study, the swollen appearance, absence of stromata and starch grains, and presence of numerous vesicles and plastoglobuli indicate severe disruption of chloroplast structure in the mutant. Therefore, the reduction in Chl content and the occurrence of aberrant structural alterations in chloroplasts should be the primary factors contributing to our study’s observed decrease in photosynthetic capacity. Comparable findings were observed in the yellow mutant of *U. pumila* [38]. As a consequence of compromised photosynthesis, mutant leaves also exhibit deficiencies in carbon skeletons and major carbon metabolism end-products, including flavonoids, leading to delayed plant growth [59].

Leaf mutations arise from a complex interaction of environmental conditions and genetic determinants [6]. Within the framework of this investigation, the simultaneous occurrence of green-leaf plants and yellow-leaf mutants suggests the importance of genetic modifications. RNA-Seq analysis is instrumental in identifying DEGs in various developmental stages and physiological conditions. Recent studies on *Hemerocallis* spp. [60] and *Primulina pungentisepala* [61] have utilized RNA-Seq technology to explore the mechanisms underlying leaf mutation. Likewise, our study utilized Illumina sequencing to identify 3024 DEGs in the yellow and green leaves. We primarily concentrated our data analysis on DEGs related to porphyrin and Chl metabolism, Car biosynthesis, photosynthesis, photosynthetic antenna proteins, and TFs that contribute to leaf coloration.

The pathway of Chl synthesis involves a cascade of enzymatic reactions catalyzed by over twenty enzymes [16]. Examining Chl precursors helps identify potential impediments in the Chl synthesis pathway [62]. In the *yl1* rice mutant, there was a significant increase in Proto IX content, while the levels of Mg-Proto IX and Pchlide were notably reduced compared to the WT. This indicates a potential impairment in the synthesis of Mg-Proto IX from Proto IX, contributing to the yellow leaf phenotype observed in the mutant [63]. Compared to wild-type plants, the levels of ALA and PBG exhibited non-significant variations, while the levels of Urogen III, Coprogen III, Proto IX, Mg-Proto IX, and Pchlide displayed a significant decrease. These findings indicated that Chl biosynthesis in the mutant should be blocked during the conversion of PBG to Urogen III in our study. Small alterations in the expression levels of the 27 genes, which encode 15 enzymes responsible for Chl synthesis in *A. thaliana*, had the potential to disturb Chl metabolism and concentration, thereby causing variations in leaf color [64]. For example, if *HEMA*, which encodes glutamyl-tRNA reductase (GluTR) responsible for catalyzing glutamyl-tRNA into glutamate-1-semialdehyde (GSA) and initiating Chl biosynthesis, is down-regulated, it can result in reduced Chl content [65,66]. The down-regulation of *HEMD*, encoding uroporphyrinogen III synthase and cyclizing and isomerizing hydroxymethylbilane to produce uroporphyrinogen III, was associated with the decrease in Chl levels observed in pink leaves compared to variegated pink-green leaves in ornamental kale [67]. Magnesium chelatase (MgCh) was crucial in regulating Chl biosynthesis through its catalytic function, inserting Mg^2+^ into protoporphyrin IX and converting it into Mg-protoporphyrin IX [68,69]. Comprising three subunits—D, I, and H—MgCh’s functionality relies on their synergistic interaction, with these subunits being conserved across plant species [6]. Reduced MgCh activity in leaf mutants was associated with a reduction in Chl content [37]. Studies have shown that suppressing the *CHLI* gene in strawberries led to yellow leaves and disrupted Chl-heme synthesis [70]. The mutant gene *GaCHLH* in *Gossypium arboreum* failed to interact effectively with *GaCHLD*, thereby hindering MgCh assembly and disrupting Chl synthesis [71]. In our investigation, the up-regulation of *CHLH* and *CHLD* may disrupted the synergy among the three subunits, ultimately leading to decreased Chl biosynthesis in the mutant. Both Chl and heme originate from a shared precursor molecule, glutamyl-tRNA^glu^, constituting tetrapyrrole molecules. Because their synthesis competes for the same substrate, changes in heme metabolism can impact the rate of Chl synthesis [16]. Feedback regulation from excessive heme inhibits ALA synthesis, disrupting Chl metabolism and causing changes in leaf color [72]. In investigating the *grc1* rice mutant, researchers identified a 45-kb insertion within the gene *LOC_Os06g40080*, encoding a heme oxygenase (*HO1*), negatively impacting Chl synthesis [73]. In a similar vein, the yellow-green leaf mutant of *B. napus* showcased decreased Chl biosynthesis and increased heme production due to an Ile320Thr mutation in *BnaC08g34840D* [74]. We found increased expression of heme synthase pathway genes (*COX10*), which had disrupted Chl metabolism in the mutant.

Changes in leaf coloration can also stem from mutations influencing the Chl degradation process. The initial step in Chl, a breakdown facilitated by Mg-dechelatase, involved the removal of the central Mg ion and was encoded by the Stay-Green (SGR) gene [75]. *Arabidopsis* mutants lacking pheophorbide oxygenase displayed stay-green leaves, while overexpression of *SGR* in rice resulted in a yellowish-brown phenotype [76]. In line with these findings, the yellow type of *I. × attenuata* ‘Sunny Foster’ showed significantly reduced Chls concentrations. Key DEGs related to Chl synthesis were suppressed, while Chl degradation was enhanced, resulting in reduced Chl accumulation and the observed lower Chls levels.

Cars, acting as accessory light-harvesting pigments, capture and transfer light energy to Chls, playing a crucial role in photoprotection within photosystems, particularly under high light conditions [77]. They contributed to the yellow-to-red hues observed in leaves [78]. Down-regulation of genes related to Car biosynthesis could lead to leaf yellowing [79]. In our analysis, we identified six DEGs associated with Car biosynthesis. Among them, four showed significant down-regulation in the mutant plant, namely *CCS*, *NCED1*, *NCED2*, and *CYP707A1*. The Car content typically experienced a less pronounced reduction than Chl content in many yellow-leaf mutants, although both notably decreased [51]. Consequently, the ratio of Car/Chl significantly increased, suggesting an elevated Car content that should contributed to the yellow leaf phenotype.

Light energy absorption occurs through pigments, initiating its transfer to the reaction centers within the thylakoid membrane [80], encompassing PSII, cytochrome f/b6 (cytf/b6), PSI, the Chl a/Chl b light-harvesting complex (LHC), and ATP synthase [81]. The reaction centers are crucial in facilitating electron transfer driven by light, absorbing and harnessing its energy [82]. In Chl-deficient rice mutants, crucial PSI core proteins (psaA and psaB), subunits of ATP synthase, cytochrome, and the LHC, were significantly decreased [83]. The down-regulation of genes involved in photosynthesis impaired the function of these complexes, hindering photosynthesis [84]. Significant alterations in the phosphorylation levels of PsbC, PsbO, PsbP, LHCA2, and LHCA3 proteins were key factors contributing to the observed photosynthetic disparities between the yellowing mutant *yl1* and the wild type [85]. Previous studies had also proposed a connection between diminished expression of photosynthesis-related DEGs, like *psbC, petD,* and *atpA*, and compromised light absorption, leading to inefficiencies in energy transfer to reaction centers [15,86]. In our study, the yellow mutant exhibited reduced expression levels of five photosynthesis-related DEGs (two *psbC*, two *petH*, one *petF*), in contrast to the wild-type plants. This observation aligned with the diminished photosynthetic capacity and abnormal chloroplasts observed in the mutant. The decreased activity of photosynthesis genes likely contributed to the noticeable decline in photosynthetic potential observed in the yellow mutant.

TFs play pivotal roles in orchestrating the precise temporal and spatial expression patterns of target genes, exerting control over their transcriptional activity through activation or repression mechanisms [87]. They exert influence or control over numerous molecular functions and biological processes, including pigment biosynthesis and chloroplast development [88,89]. The differentially expressed TFs identified in our study belonged to gene families such as *bHLH*s, *NAC*s, *MYB-related*, *ERF*s, *B3*s, *C2H2*s, *GRAS*s, *LBD*s, *MYB*s, *WRKY*s, and *GLK*s. Similar findings were reported in a yellow-leaf mutant of *G. biloba* [28]. The suppression of *bHLH71*-*like* in pepper plants led to elevated levels of zeaxanthin and antheraxanthin, implying a role for *bHLH71*-*like* as a promoter of Car biosynthesis in the *yl1* yellow pepper mutant [90]. Moreover, *bHLH*s also regulate the formation and development of stomata. Knocking out the *bHLH*-type TFs *OsICE1* and *OsICE2* in *A. thaliana* resulted in the loss of stomata [91]. Studies have demonstrated that *ANAC046* positively regulates Chl biosynthesis in *A. thaliana*, interacting with the promoter regions of key genes involved in Chl synthesis, including *NYC1*, *SGR1*, *SGR2*, and *PAO* [92]. Gene families like *MYB-related* [93], *ERF*s [94], *WRKY*s [95], and *MYB*s [96] have been documented to exert significant influence on Chl metabolism and consequently impact leaf coloration. The *GLK* TFs, found within the newly classified GARP superfamily in plants, were known to participate in a range of biological processes, including the formation of chloroplasts [97,98]. In flowering plants, a typical occurrence involved a pair of *GLK* genes, namely *GLK1* and *GLK2*, expressed in photosynthetic tissues and performing overlapping functions [98]. In rice, enhancing chloroplast formation in both callus and roots had been observed through the overexpression of *OsGLK1* [99]. In *Arabidopsis*, *Atglk* double mutants displayed pale green leaves, with chloroplasts measuring approximately half the size in the cross-sectional area compared to wild-type ones [100]. Research suggested that *BpGLK1* is critical in regulating chloroplast development in the yellow-green mutant by binding directly to the promoters of genes linked to *BpCHLH* and *BpCRD1* [1]. Overexpression of *MdGLK1* in the *MdGLK* double mutant alleviated the abnormalities in chloroplast structure and function, leading to the up-regulation of genes associated with Chl biosynthesis (*CHLH*, *CAO*) and photosynthesis (*LHCB1-5*) [101]. Similarly, lower expression of *GLK*s had been observed in yellow leaves of *G. biloba* compared to green leaves [15]. In our investigation, seven *GLK* TFs, including *GLK1* and *GLK2*, showed decreased expression levels in the yellow mutant, underscoring a tight association between changes in leaf color and chloroplast development. These discoveries provide further insights into the involvement of TFs’ role in modulating pigment metabolism, chloroplast maturation and influencing leaf coloration in *I. × attenuata* ‘Sunny Foster’.

## 4. Conclusions

This study thoroughly examined cytological, physiological, and transcriptomic variances between wild-type and yellow mutant leaves. The yellow leaf mutant exhibited aberrant chloroplast morphology, notable discrepancies in pigment levels, limited crucial Chl precursors synthesis, and diminished photosynthetic efficiency compared to its wild type. The decrease in Chl b content exceeded that of Chl a, classifying this mutant as Chl b-deficient. Transcriptome analysis elucidated a spectrum of DEGs linked to pigment metabolism in the yellow mutant. The decreased expression of *CHLD* and *CHLI*, indicating unsynergistic interactions with *CHLH*, as well as the diminished expression of *HEMD* and the heightened expression of *COX10*, disrupted Chl synthesis while favoring heme synthesis.

Furthermore, the Chl degradation process was heightened by the increased expression of *SGR*. As a result, these combined effects significantly reduced Chl concentration within the mutant. Additionally, a reduction in the activity of genes involved in Car biosynthesis was noted, corresponding with the lower carotenoid content. The down-regulation of TFs such as *GLK1*, *GLK2*, and *bHLH*s contributed to the defective chloroplasts. These effects ultimately culminated in a decrease in Chl content and an elevation in the Car/Chl ratios, manifesting as the yellow leaf phenotype (Figure 10). In conclusion, our findings offer meaningful gene resources that could inform plant breeding efforts to enhance leaf color traits.

## 5. Materials and Methods

### 5.1. Plant Materials

The study analyzed green leaves from the cultivar *I.* × *attenuata* ‘Foster #2’ and yellow leaves from its naturally occurring bud mutant cultivar, ‘Sunny Foster’, characterized by the presence of newly emerged yellow foliage, as shown in Figure 1. Formal identification of these plant materials was conducted by Professor Donglin Zhang from the University of Georgia. Voucher specimens of these plant materials have been deposited in the Nanjing Forestry University *Ilex* Germplasm Resource Bank with the deposit numbers DI00025 and DI00026. Both varieties were 2-year-old cutting liners, planted in one-gallon plastic containers. Each pot was filled with a substrate composed of a 3:1 volume ratio mixture of peat and perlite, with the substrate pH value set at 6.5. A cohort of 200 two-year-old plants constituted the experimental population. These specimens were grown under standardized environmental conditions at Nanjing Qingzhuguo Landscape Horticulture Co., Ltd. (Nanjing, China).

On a cloudless day in June 2023, green leaves from the WT cultivar and yellow leaves from the YT cultivar were collected. Three leaves were collected from each individual plant of both varieties. For each biological replicate, three randomly selected plants from both the wild type and mutant type were utilized. Three measurements were conducted for each accession to ensure robustness of the data. Following collection, the harvested leaves were immediately flash-frozen in liquid nitrogen and stored at −80 °C for subsequent analyses. All leaf samples were collected with the approval of the owner of Nanjing Qingzhuguo Landscape Horticulture Co., Ltd. (Nanjing, China).

### 5.2. Color Indice Measurement

A handheld precision colorimeter (CR-8, Shenzhen 3nh Technology Co., Ltd. Shenzhen, China) was employed to record three color parameters: L*, a*, b* values. Following this, the hue angle (H° = arctan (b*/a*), color ratio (a*/b*), and chroma (C = (a*^2^ + b*^2^)^1/2^) were computed based on previously reported formulas [102]. The parameter L* indicates brightness, measured on a scale ranging from 0 to 100, while a* and b* represent chromaticity without predefined numerical boundaries. Negative values of a* indicate green hues, while positive values suggest red. Similarly, negative b* values correspond to blue, and positive values signify yellow. The parameter C quantifies color intensity. The hue angle (H°) is utilized to characterize colors, with red associated with a hue angle of 0°, yellow with 90°, green with 180°, and blue with 360° [103]. Three separate plants were measured for each variety, with three random points selected on each leaf from the upper, middle, and lower regions and their average values were calculated.

### 5.3. SPAD, Pigment and Chl Precursors Content Measurements

The SPAD value, serving as an indicator of Chl content, was determined utilizing a SPAD-502 Chl meter (SPAD, Konica Minolta Sensing, Tokyo, Japan). For Chl extraction, 0.2 g of both green and yellow leaves were finely chopped and immersed in 95% ethanol overnight. Subsequently, the extracted Chl and Car were quantified at wavelengths of 470 nm, 649 nm, and 665 nm using a UV-1800 spectrophotometer (DU-800, Beckman Coulter, Brea, CA, USA), following the method described by Lichtenthaler [104]. Anthocyanins were extracted in a solution consisting of methanol and HCl (99:1, *v/v*) at 4 °C in a light-free environment for 24 h, as outlined by Hughes et al. and Manetas et al. [105,106]. The anthocyanin content was determined by measuring the absorbance at 530 nm, with cyanidin-3-glucoside as the calibration standard. Flavonoid content was assessed by referencing a previous method [107]. Briefly, 0.5 mL aliquots of properly diluted extracts or standard solutions were dispensed into 15 mL conical tubes filled with 2 mL of double-distilled water. Afterward, 0.15 mL of a 5% solution of sodium nitrite (NaNO_2_) was incorporated, allowing for a 5 min reaction period. This was followed by the addition of 0.15 mL of 10% solution of aluminum chloride hexahydrate (AlCl_3_·6H_2_O), followed by an additional 5 min incubation period. After adding 1 mL of a 1 M sodium hydroxide (NaOH) solution, the resulting mixture was vigorously agitated and then incubated for 15 min. The absorbance of the solution was measured at a wavelength of 415 nm.

The procedure involved pulverizing 50 mg of fresh leaves into a fine powder using liquid nitrogen. The powder was combined with 450 mL of 0.01 mol L^−1^ PBS solution (pH 7.4) and subjected to centrifugation at 12,000× *g* for 10 min to obtain the supernatant. Plant Enzyme-Linked ImmunoSorbent Assay kits (#YJ503247, Yuanju, Shanghai, China) were utilized to quantify the levels of seven Chl precursor compounds, including ALA, PBG, Urogen III, Coprogen III, Proto IX, Mg-Proto IX, and Pchlide, in both types of leaves.

### 5.4. Leaf Anatomical Structure Observation

Leaf samples were sliced into segments measuring 1 cm × 2 cm and immersed in a fixative solution comprising formalin, alcohol, and glacial acetic acid (in a ratio of 90:5:5 by volume) for a minimum of 48 h. After fixation, the leaves underwent dehydration using a series of ethanol solutions, followed by two xylene rinses, and subsequently encased within paraffin wax, as previously outlined [108]. Samples were permeated and paraffin-embedded. These samples were double-stained with safranin and fast green counterstain. Photomicrographs were captured using an optical microscope (BX53, Olympus, Tokyo, Japan).

### 5.5. Leaf Epidermis Microstructure Observation

Cryogenic Scanning Electron Microscopy (CryoSEM) was used to analyze leaf epidermis microstructure as detailed previously [109]. Fresh leaf fragments with 1 cm × 1 cm were firmly affixed onto 2 cm × 4 cm copper plates using computer thermal compound paste. The samples were initially frozen at −20 °C on a large metal holder before being coated with a layer of gold in an argon environment. The plate containing either the fresh, non-metalized sample or the frozen, metalized sample was securely attached to the cooling stage of a Deben Coolstage refrigerating unit (Deben, Suffolk, UK) set at −30 °C. These specimens were then examined using a Quanta 200 scanning electron microscope (FEI, Hillsboro, OR, USA) in high vacuum mode at 20 kV, with a backscattered electron detector QBSD and an 8–12 mm working distance.

### 5.6. Chloroplast Ultrastructure Observation

Following the previously described methodology [110], 1 mm × 2 mm segments were prepared for ultrastructural examination and fixed with 2.5% glutaraldehyde in 0.1 mol L^−1^ phosphate buffer (with a pH of 7.2). After three 15 min washes in phosphate buffer, they underwent a 2 h post fixation with 1% OsO_4_ in phosphate buffer (with a pH of 7.0). After three additional washes in phosphate buffer, the samples were dehydrated using a gradient series of ethanol (30%, 50%, 60%, 70%, 80%, 90%, and 95%), with each step lasting approximately 15 to 20 min. Subsequently, they were immersed in pure acetone for a duration of 1 h, with a total of seven washes. For infiltration, the samples underwent a series of steps: they were initially immersed in a 1:1 mixture of pure acetone and the final spurr resin mixture for 1 h at room temperature. Subsequently, they were transferred to a 1:3 mixture of pure acetone and the final resin mixture for 3 h. Finally, they were placed in the pure spurr resin mixture overnight. The embedded sample was then sectioned into thin slices (50–80 nm) using a diamond knife and an ultra-microtome (UC6/FC6, Leica, Wetzlar, Germany). These sections underwent staining with lead acetate and uranium dioxide for 15 min each. Observation and photography of the ultrastructure of chloroplasts were performed using a JEM 1400 transmission electron microscope (JEOL, Tokyo, Japan).

### 5.7. Photosynthetic Parameters Measurements

Between 09:00 and 11:00, a portable photosynthesis system (LI-6400, LICOR, Lincoln, NE, USA) was utilized to measure the net photosynthetic rate (P_n_), stomatal conductance (G_s_), intercellular CO_2_ concentration (C_i_), and transpiration rate (E). During the measurements, the CO_2_ concentration was held at 400 μmol mol^−1^, the leaf chamber temperature was set at 25 °C, and the light density in the sample chamber was set to 1000 μmol m^−2^ s^−1^ in the sample chamber.

On the same day, between 10:00 and 12:00, Chl fluorescence characteristics were evaluated using a pulse amplitude modulation (PAM) fluorometer (Junior-PAM-II, Heinz Walz GmbH, Effeltrich, Germany), following the procedure described previously [111]. Chl fluorescence parameters were recorded following a previously published method [24]. Before measurements, the leaves were subjected to a 30 min dark adaptation period. The photosynthetically active radiation (PAR) was maintained at 125 μmol m^−2^ s^−1^. To ensure that the Ft fluorescence level remained between 200 and 400 counts, adjustments were made to the measuring light intensity, ranging from 7 to 11, with a frequency set at 1 or 2. Using the WinControl-3 software (version 3.32), parameters such as fluorescence origin (Fo), fluorescence maximum (Fm) and variable fluorescence (Fv = Fm − Fo), the maximum quantum yield of photosystem II (Fv/Fm), actual photosynthetic efficiency of photosystem II (ΦPSII), non-photochemical quenching (NPQ), regulatory energy dissipation quantum yield (Y (NPQ)), and non-regulatory energy dissipation quantum yield (Y (NO)) were automatically captured and calculated. Each biological replicate consisted of three leaves sampled from individual plants of each variety, with three randomly selected plants from both the wild and mutant types.

### 5.8. Plant Height, Biomass, Soluble Sugar, and Soluble Protein Measurements

Plant height for each variety was measured in triplicate, with nine plants in each replicate. From each group, three randomly selected plants were excavated, washed with deionized water, and wiped clean to determine both fresh and dry mass. The fresh mass of each plant was recorded individually before being subjected to drying in an oven (Model 202-0B, Shangyi, Shanghai, China) at 65 °C for 12 h to obtain the dry mass for each treatment. Soluble sugar content was assessed via anthrone colorimetry [112], whereas quantification of soluble protein content followed the Bradford method [113], with bovine serum albumin serving as the standard.

### 5.9. RNA Extraction and Preparation of cDNA Library

Total RNA extraction from the green leaves of WT and the yellow leaves of YT was carried out using the NEBNext^®^ UltraTM Plant RNA Extraction Kit (TaKaRa, Dalian, China). Each RNA sample underwent three biological replicates. Evaluation of RNA sample quantity and quality involved 1% agarose gel electrophoresis and a Nanodrop 1000 spectrophotometer (Nanodrop, Wilmington, DE, USA). Assessment of RNA integrity and concentration was performed using an Agilent 2100 Bioanalyzer (Agilent Technologies, Inc., Santa Clara, CA, USA). Subsequently, mRNA isolation was conducted using the NEBNext Poly (A) mRNA Magnetic Isolation Module (E7490; NEB, Ipswich, MA, USA), followed by fragmentation into 200 nt short RNA inserts. The fragmented mRNA underwent first and second-strand cDNA synthesis. Following end-repair, dA-tailing, and adaptor ligation on the double-stranded cDNA, PCR amplification was performed to enrich appropriate fragments.

### 5.10. Illumina Deep Sequencing and Data Analysis

Sequencing of the cDNA libraries obtained from the samples was conducted on an Illumina NovaSeq 6000 platform (Illumina, San Diego, CA, USA). Processing the raw data (raw reads) involved the utilization of custom Perl scripts to filter out adaptors, N bases, and low-quality reads, resulting in the acquisition of clean reads. Quality metrics such as Q20, Q30, and GC content were assessed for the clean data. Following this, alignment of the clean reads to the *I. latifolia* reference genome (https://ngdc.cncb.ac.cn/gwh, accessed on 1 December 2023, GWHBIST00000000) was performed using HISAT software (version 2.0.4) with default parameters. Calculation of read counts for each gene was accomplished using HTSeq (version 0.6.1), and gene expression levels were quantified in terms of fragments per transcript kilobase per million fragments mapped (FPKM) values.

### 5.11. Identification and Functional Analysis of DEGs

DESeq2 R software (version 1.20.0) was employed to detect DEGs in libraries derived from both green and yellow leaves. This software offers statistical procedures tailored for digital gene expression data analysis using a negative binomial distribution model. To control false discovery rates, the resulting *p*-values underwent adjustment using Benjamini and Hochberg’s method. Genes with a *Padj* ≤ 0.05 and |log_2_(Fold Change)| ≥ 1 were deemed significantly differentially expressed. Additionally, we utilized the cluster Profiler R package (Version 3.5.0) to conduct enrichment analyses for Gene Ontology (GO) and Kyoto Encyclopedia of Genes and Genomes (KEGG) pathways. Gene Set Enrichment Analysis (GSEA) was performed to assess whether predefined gene sets exhibited significant and consistent differences between the two biological states. For GSEA, the local GSEA analysis tool (http://www.broadinstitute.org/gsea/index.jsp, accessed on 5 December 2023) was utilized with the KEGG dataset. Enrichment analysis revealed significant enrichment of all DEG sets, including both up-regulated and down-regulated DEGs, across all differential comparison combinations.

### 5.12. qRT-PCR Verification

Gene-specific primers were designed with Primer Premier 6.0 (Premier Biosoft Inc., Palo Alto, CA, USA), utilizing the *Actin* gene as the internal reference [114]. Total RNA was carried out using the RNAprep Pure Plant Plus Kit (#DP432, Tiangen, Beijing, China), followed by cDNA synthesis using the HiScript III 1st Strand cDNA Synthesis Kit (#R312-01, Vazyme, Nanjing, China). qRT-PCR was performed using AceQ Universal SYBR qPCR Master Mix (#Q511-02, Vazyme, Nanjing, China). Each sample underwent technical triplicates with three biological replicates. Relative expression levels of target genes were assessed using the 2^−ΔΔCt^ comparative Ct method [115].

### 5.13. Statistical Analysis

GraphPad Prism 9.0 software (GraphPad Inc., La Jolla, CA, USA) and SPSS 19.0 software (SPSS Inc., Chicago, IL, USA) were employed for statistical analysis. Significance levels were determined via the *t*-test, where significance was denoted by asterisks as follows: **** *p* < 0.0001, *** *p* < 0.001, ** *p* < 0.01, * *p* < 0.05, while non-significant results were labeled as “ns”. Error bars represent the standard deviation of the mean (SD). Image analyses were carried out using ImageJ (https://imagej.net/ij/, accessed on 15 December 2023). For cytological images, 10 randomly selected views were analyzed.

## Figures and Tables

**Figure 1 plants-13-01284-f001:**
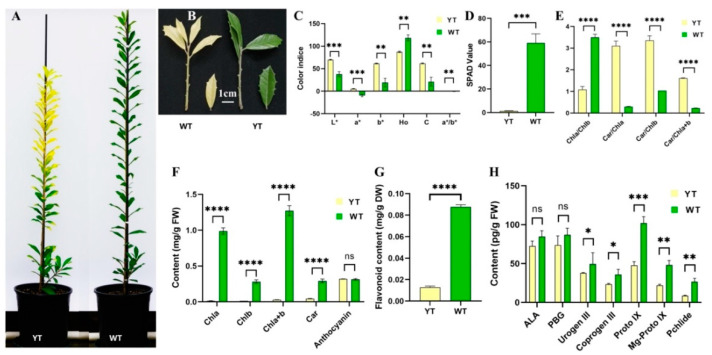
Leaf phenotypes, color indices, pigments, and Chl precursors content of YT and WT. (**A**) YT and WT plants. (**B**) Leaf phenotypes of YT and WT. Scale bar = 1 cm. (**C**) Leaf color indices of YT and WT. (**D**) SPAD value of YT and WT. (**E**) Pigments ratios of YT and WT. (**F**) Pigments content of YT and WT. (**G**) Flavonoid content of YT and WT. (**H**) Chl precursors content of YT and WT. The values are presented as mean ± SD. Statistical significance was determined as follows: ns (not significant), * *p* < 0.05, ** *p* < 0.01, *** *p* < 0.001, **** *p* < 0.0001, using the *t*-test. ALA—5-aminolevulinic acid; Car—carotenoids; Chl—chlorophyll; Coprogen III—coproporphyrinogen; FW—fresh weight; Mg-Proto IX—Mg-protoporphyrin IX; PBG—porphobilinogen; Pchlide—protochlorophyllide; Proto IX—protoporphyrin IX; SD—standard deviation; SPAD—Soil and Plant Analyzer Development; Urogen III—uroporphyrinogen III; WT—wild-type *I.* × *attenuata* ‘Foster #2’; YT—mutant *I.* × *attenuata* ‘Sunny Foster’.

**Figure 2 plants-13-01284-f002:**
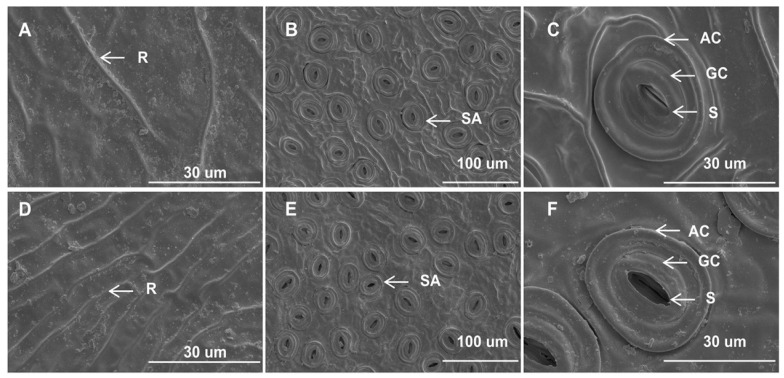
Epidermal features of YT and WT. (**A**–**C**) are YTs. (**D**–**F**) are WTs. (**A**) The upper epidermis of YT. (**D**) The upper epidermis of WT. (**B**,**C**) Lower epidermis of YT. (**E**,**F**) Lower epidermis of WT. Scale bars = 100 μm (**B**,**E**), 30 μm (**A**,**C**,**D**,**F**). AC—arch cover; GC—guard cell; R—ridge; SA—stomatal apparatus; S—stoma; WT—wild-type *Ilex* × *attenuata* ‘Foster #2’; YT—mutant *I.* × *attenuata* ‘Sunny Foster’.

**Figure 3 plants-13-01284-f003:**
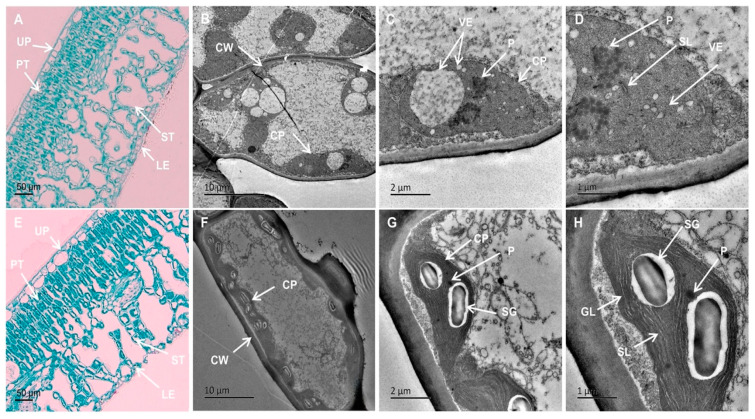
Anatomical structures and chloroplast ultrastructures of YT and WT. (**A**–**D**) are YTs. (**E**–**H**) are WTs. Scale bars = 50 μm (**A**,**E**), 10 μm (**B**,**F**), 2 μm (**C**,**G**), 1 μm (**D**,**H**). CP—chloroplast; CW—cell wall; GL—grana lamella; LE—lower epidermis; P—Plastoglobuli; PT—palisade tissue; SG—starch grain; SL—stroma lamella; ST—spongy tissue; UP—upper epidermis; VE—vesicle; WT—wild-type *Ilex* × *attenuata* ‘Foster #2’; YT—mutant *I.* × *attenuata* ‘Sunny Foster’.

**Figure 4 plants-13-01284-f004:**
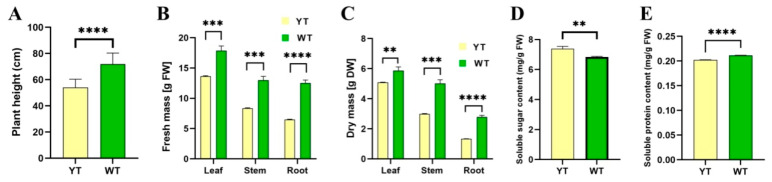
Plant height, fresh mass, dry mass, and soluble sugar and protein content of YT and WT. (**A**) Plant height of YT and WT. (**B**) Fresh mass of YT and WT. (**C**) Dry mass of YT and WT. (**D**) Soluble sugar content of YT and WT. (**E**) Soluble protein content of YT and WT. The values are presented as mean ± SD. Statistical significance was determined as follows: ** *p* < 0.01, *** *p* < 0.001, **** *p* < 0.0001, using the *t*-test. DW—dry weight; FW—fresh weight; SD—standard deviation; WT—wild-type *Ilex* × *attenuata* ‘Foster #2’; YT—mutant *I.* × *attenuata* ‘Sunny Foster’.

**Figure 5 plants-13-01284-f005:**
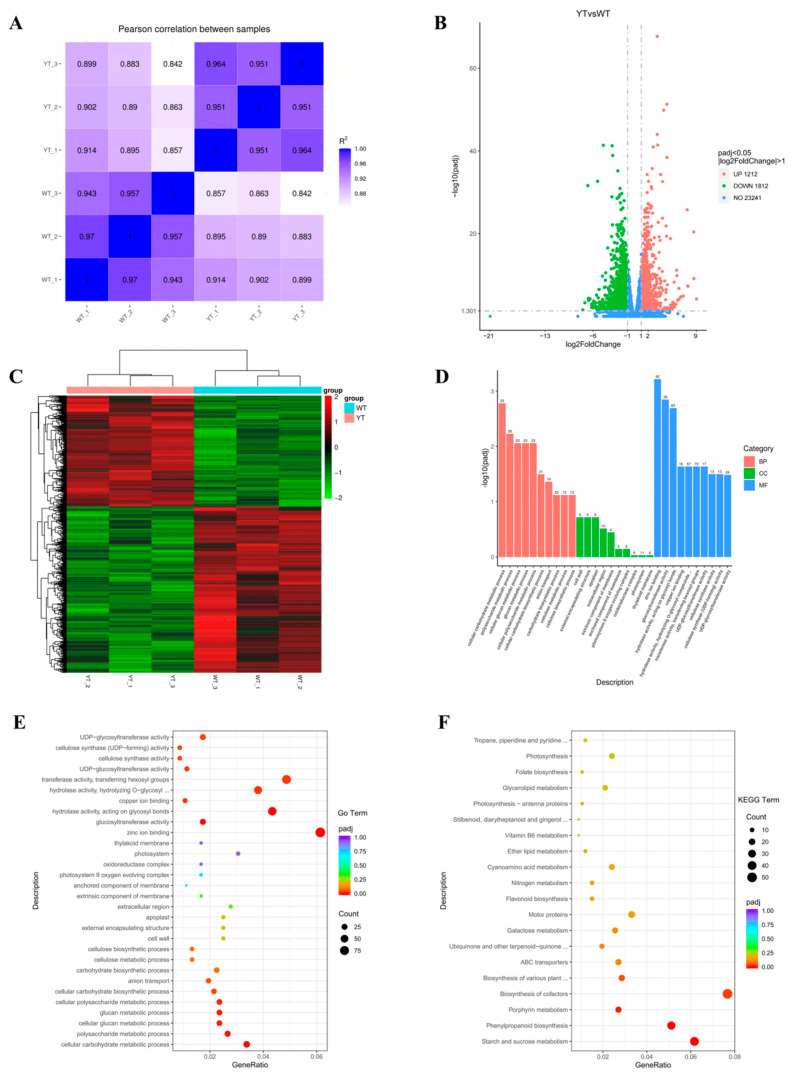
GO and KEGG enrichment of DEGs between YT and WT. (**A**) Pearson correlation between different samples. (**B**) DEGs were visualized using a volcano plot. On the *x*-axis, log_2_ (fold change) is represented, while the *y*-axis depicts −log_10_*Padjj* values. Up-regulated genes are denoted by red points, whereas down-regulated genes are indicated by green points. (**C**) Clustering of genes between different samples. Red denotes genes that are up-regulated, and green indicates genes that are down-regulated. (**D**) GO analyses of the DEGs. (**E**) The 30 most enriched enriched GO terms. (**F**) The 20 most enriched KEGG terms. The *x*-axis illustrates the enrichment score, while the size of the bubbles corresponds to the number of DEGs. Bubble color transitions from purple to red indicate decreasing *Padjs* (increasing significance). WT—wild-type *Ilex* × *attenuata* ‘Foster #2’; YT—mutant *I.* × *attenuata* ‘Sunny Foster’.

**Figure 6 plants-13-01284-f006:**
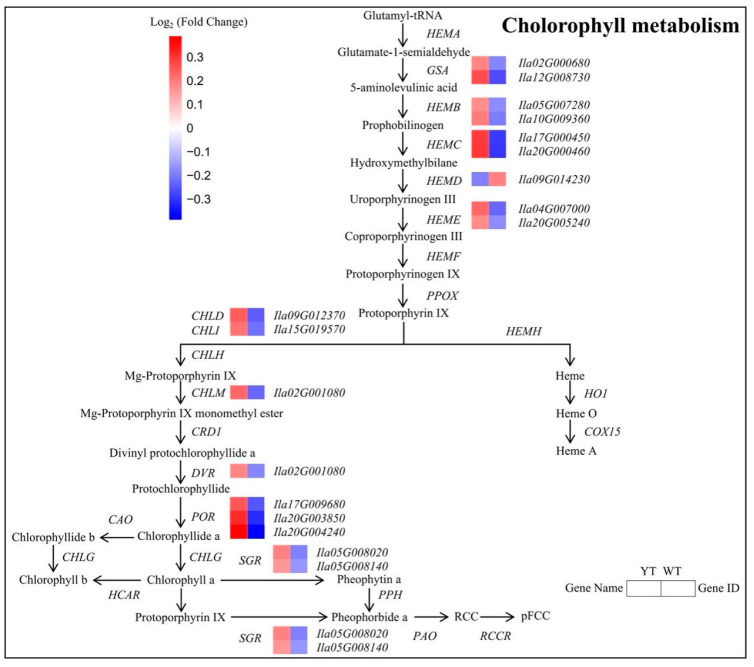
DEGs involved in the chlorophyll metabolism pathway between YT and WT. The mean log_2_ (Fold Change) values for the DEGs were computed based on three biological replicates for YT and WT. Up-regulated genes are denoted in red, while down-regulated genes are shown in blue. WT—wild-type *Ilex* × *attenuata* ‘Foster #2’; YT—mutant *I.* × *attenuata* ‘Sunny Foster’.

**Figure 7 plants-13-01284-f007:**
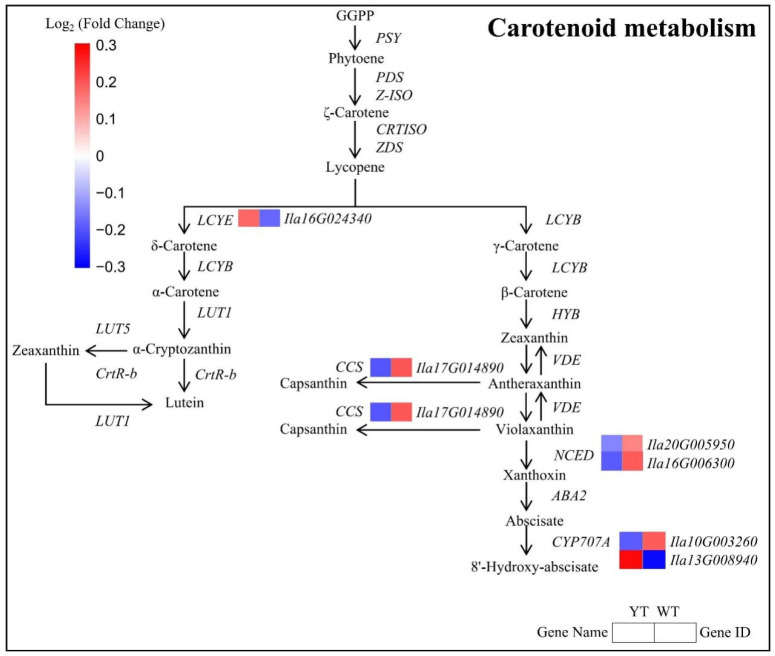
DEGs involved in the carotenoid metabolism pathway between YT and WT. The mean log_2_ (Fold Change) values for the DEGs were computed based on three biological replicates for YT and WT. Up-regulated genes are denoted in red, while down-regulated genes are shown in blue. WT—wild-type *Ilex* × *attenuata* ‘Foster #2’; YT—mutant *I.* × *attenuata* ‘Sunny Foster’.

**Figure 8 plants-13-01284-f008:**
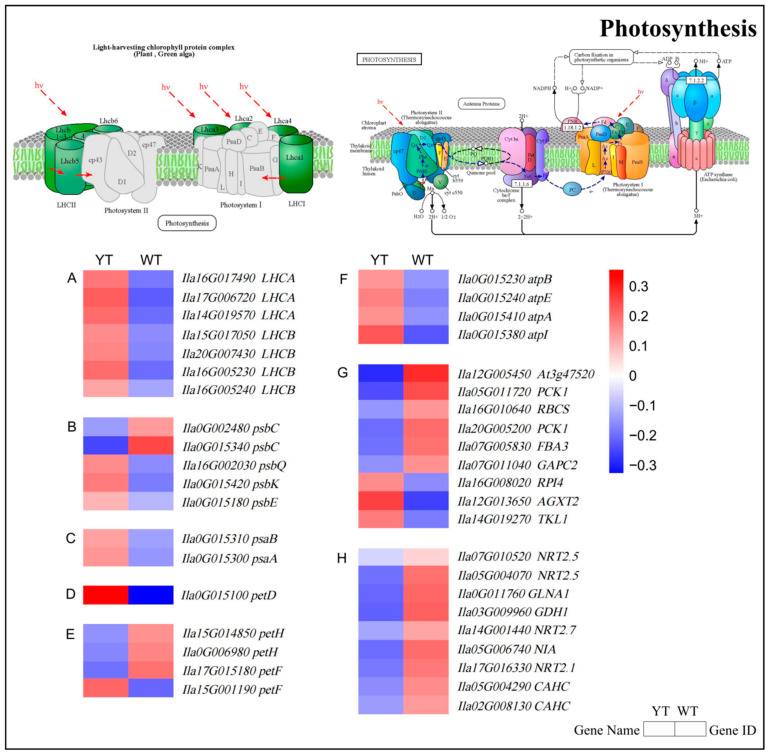
The expression profiles of DEGs involved in photosynthesis and nitrogen metabolism between YT and WT. (**A**) DEGs involved in photosynthesis-antenna protein. (**B**) DEGs involved in photosystem II. (**C**) DEGs involved in photosystem I. (**D**) DEGs involved in cytochrome b6/f complex. (**E**) DEGs involved in photosynthetic electron transport. (**F**) DEGs involved in F-type ATPase. (**G**) DEGs involved in carbon fixation in photosynthetic organisms. (**H**) DEGs involved in nitrogen metabolism. The mean log_2_ (Fold Change) values for the DEGs were computed based on three biological replicates for YT and WT. Up-regulated genes are denoted in red, while down-regulated genes are shown in blue. WT—wild-type *Ilex* × *attenuata* ‘Foster #2’; YT—mutant *I.* × *attenuata* ‘Sunny Foster’. The light-harvesting chlorophyll protein complex schematic in the top-left corner and the photosynthesis pathway schematic in the top-right corner are sourced from the KEGG pathway website (https://www.kegg.jp, accessed on 22 December 2023).

**Figure 9 plants-13-01284-f009:**
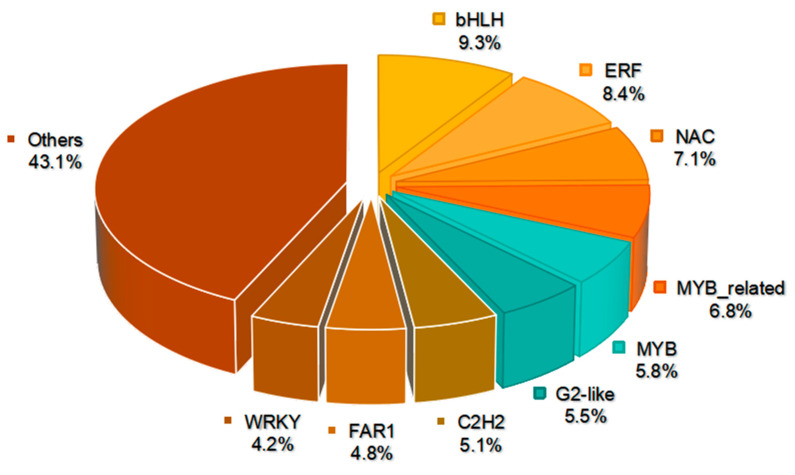
DEGs involved in transcription factors between YT and WT.

**Figure 10 plants-13-01284-f010:**
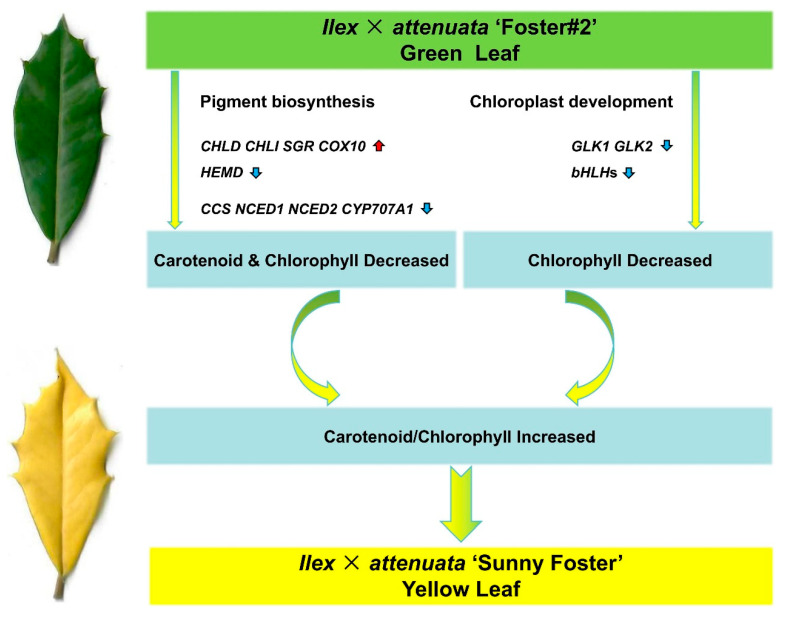
A proposed model for potential molecular leaf mutations in *I.* × *attenuata* ‘Sunny Foster’. Gene upregulation is indicated by the vertical red arrow; gene downregulation is denoted by the vertical blue arrow. The green leaf originates from the wild-type plant (*I.* × *attenuata* ‘Foster#2’), and the yellow leaf is derived from the mutant plant (*I.* × *attenuata* ‘Sunny Foster’).

**Table 1 plants-13-01284-t001:** Comparative analysis of chloroplast ultrastructure of YT and WT.

Ultrastructural Parameters	YT	WT
Chloroplast number (per cell)	3.90 ± 0.74 ^b^	9.60 ± 0.84 ^a^
Chloroplast length (um)	6.46 ± 0.86 ^a^	5.98 ± 0.80 ^a^
Chloroplast width (um)	5.34 ± 1.16 ^a^	2.31 ± 0.48 ^b^
Chloroplast length/width	1.25 ± 0.29 ^b^	2.67 ± 0.52 ^a^
Starch grain number (per cell)	0.00 ± 0.00 ^b^	2.10 ± 0.57 ^a^

Note: The data are expressed as means ± SD (standard deviation) from three independent biological replicates. Statistical significance was determined at *p* < 0.05, denoted by different lowercase letters. WT—wild-type *Ilex* × *attenuata* ‘Foster #2’; YT—mutant *I.* × *attenuata* ‘Sunny Foster’.

**Table 2 plants-13-01284-t002:** Photosynthetic parameters in the leaves of YT and WT.

	Parameters	YT	WT
Photosynthesis	P_n_ [μmol(CO_2_) m^−2^ s^−1^]	0.37 ± 0.55 ^b^	19.47 ± 2.14 ^a^
	E [mol(H_2_O) m^−2^ s^−1^]	0.87 ± 0.21 ^b^	4.33 ± 0.38 ^a^
	G_s_ [mol(H_2_O) m^−2^ s^−1^]	40.67 ± 8.33 ^b^	296.00 ± 43.35 ^a^
	C_i_ [μmol(CO_2_) mol^−1^]	428.00 ± 26.00 ^a^	274.67 ± 5.13 ^b^
Chl fluorescence	Fo	153.00 ± 13.00 ^b^	217.33 ± 24.58 ^a^
	Fm	184.33 ± 13.80 ^b^	762.00 ± 61.88 ^a^
	Fv	31.33 ± 4.62 ^b^	544.67 ± 37.63 ^a^
	Fv/Fm	0.17 ± 0.03 ^b^	0.72 ± 0.01 ^a^
	ΦPSII	0.11 ± 0.05 ^b^	0.48 ± 0.01 ^a^
	NPQ	0.02 ± 0.02 ^b^	0.48 ± 0.11 ^a^
	Y (NO)	0.87 ± 0.04 ^a^	0.35 ± 0.02 ^b^
	Y (NPQ)	0.02 ± 0.02 ^b^	0.17 ± 0.03 ^a^

Note: The data are expressed as means ± SD (standard deviation) from three independent biological replicates. Statistical significance was determined at *p* < 0.05, denoted by different lowercase letters. C_i_—intercellular CO_2_ concentration; E—transpiration rate; Fo—fluorescence origin; Fm—fluorescence maximum; Fv—variable fluorescence; Fv/Fm—maximal quantum yield of PSII photochemistry; G_s_—stomatal conductance; NPQ—non-photochemical quenching; P_n_—net photosynthetic rate; WT—wild-type *Ilex* × *attenuata* ‘Foster #2’; Y (NO)—non-regulatory energy dissipation quantum yield; Y (NPQ)—regulatory energy dissipation quantum yield; YT—mutant *I.* × *attenuata* ‘Sunny Foster’; ΦPSII—actual photosynthetic efficiency of PSII.

## Data Availability

All Illumina sequencing data can be found at the NCBI Sequence Read Archive (NCBI SRA) under BioProject accession number PRJNA1035424. Data will be made available on request.

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
