# Peer review of "Molecular Mechanisms of Chlorophyll Deficiency in Ilex × attenuata ‘Sunny Foster’ Mutant"

_plants, 2024, doi:10.3390/plants13101284_

Round 1

Reviewer 1 Report

Comments and Suggestions for Authors

The objective of the current study conducted by Chibuogwu et al.  entitled “Molecular Mechanisms of Chlorophyll Deficiency in Ilex × attenuata ‘Sunny Foster’ Mutant is very similar to previously reported paper Zou et al.; in Molecular insights into a non-lethal yellow bud mutant in Ilex × ‘Nellie. Scientia Horticulturae 329 (2024) 113033.

Based on this motif, I suggest rejecting this manuscript

e.g.  in this abstract: “Through cytological, physiological, and transcriptomic analyses, we identified remarkable differences between the mutant and its wild type. The mutant exhibited abnormal chloroplasts, reduced chlorophyll levels, elevated carotenoid/chlorophyll ratios, and delayed plant growth.”

In the abstract of the paper ..Zou et al.; in Molecular insights into a non-lethal yellow bud mutant in Ilex × ‘Nellie. Scientia Horticulturae 329 (2024) 113033.

“Through cytological, physiological, metabolomic, and transcriptomic analyses, we identified remarkable differences between the mutant and its wild-type. The mutant exhibited abnormal chloroplasts, reduced chlorophyll levels, and elevated carotenoid/chlorophyll ratios.”

Reviewer 2 Report

Comments and Suggestions for Authors

In this manuscript, the authors investigate the pigment contents and differential gene expression of the ‘Sunny Foster’ cultivar of Ilex x attentata, providing valuable insight into potential gene targets for leaf color enhancement. Clarification of a few points would improve the manuscript:

Line 132 – This states that the plants were cultivated under “similar” environmental conditions. Similar to a set of environmental conditions stated elsewhere, or the environmental conditions for the WT and YT plants were similar to each other? Please clarify.

Figure 4 – Part of the y-axis label for panel D is cropped; please replace this with an image that includes the entire text.

Figure 8 – Please include the source(s) of the thylakoid diagrams, if you did not create them yourselves.

Comments on the Quality of English Language

Minor English language edits would be helpful. 

Reviewer 3 Report

Comments and Suggestions for Authors

This manuscript reviews the physiological, morphological and molecular differences between wild type and yellow leaf mutant derived from I. × attenuata ‘Foster#2’. This is an interesting work that was carried out using various research methods. However, there are some questions and small comments to the manuscript.

1. It is not clear why, with such an extreme drop in chlorophyll and photosynthesis rates, plant growth and biomass did not decrease so significantly. There are probably compensation mechanisms, please highlight what that might be.

2. It remains completely unclear why most of the expression profiles of DEGs involved in photosynthesis are higher in the mutant.

3. It is recommended to provide the primary parameters of chlorophyll fluorescence, such as Fo, Fmax and Fv. This will expand our understanding of how the photosystem II works in the mutant. Since the parameters Fv/Fm and others are recalculated per one unit of photosystems.

4. The designation Y (II) is obsolete; the designation ФPSII is now used in photobiology.

5. The arrangement of wild-type and mutant indicators should be unified. Now in most figures the wild type is on the left, but in fugures 7 and 8 it is on the right.

6. Figure 5 is too small. Arrange its parts differently or break it into several figures.

Reviewer 4 Report

Comments and Suggestions for Authors

The manuscript entitled “Molecular Mechanisms of Chlorophyll Deficiency in Ilex × attenuata ‘Sunny Foster’ Mutant” used cytological, physiological, and transcriptomic analyses and identified remarkable differences between the mutant and its wild type. This research has great scientific interest and could accept the manuscript in its present form. The article is well written, and I would like to recommend it for publication. However, it would be better if you could include some more information on the connection between Gs, and Ci in WT and YT (Table 2) for color reflection in the relevant sections (Results and Discussion). It would be better if you could recheck for minor typos and italicized Latin names throughout the manuscript.

Reviewer 5 Report

Comments and Suggestions for Authors

The regular paper of Zou et al. reveals a comparative analysis between WT (I. × attenuata ‘Foster#2’) and YT mutant (I. × attenuata ‘Sunny Foster’) defective in Chlorophyll biosynthesis, with a higher extent deficiency in Chl b, which displays et yellowish leaves, lower fresh and dry biomass, and shorter plants compared to WT. It seems this mutation disturbed almost all the Chlorophyll fluorescence- and photosynthesis-related parameters. The authors assessed physiological, cytological and transcriptomic parameters. They also delved into tranccriptomics to pinpoint the potential genes involved in Chl biosynthesis and responsible for the difference in leaf color between the two ecotypes (WT and YT). They successfully identified some interesting genes and intriguingly their transcriptomic results were validated by qRT-PCR analysis, which may consolidate their RNA-Seq data. The results were consistent and coherent. Manuscript is also well written and results well presented. So, I recommend checking these minor points below, before being considered for publication.

Major points:

- Line 39: you wrote «These mutants significantly affect pigment synthesis…… »

Revisions: Sentence mal-constructed. Write as follows:  These mutations significantly affect pigment synthesis…., or these mutants defective in pigment synthesis…….

- Line 345: you wrote «were largely closed..»

Please correct as follows: were almost fully (or completely) closed.

- Line 417-418: you wrote «Moreover, the levels of soluble sugar and soluble proteinTh in mutant plant leaves significantly decreased by 7.45% and 4.76%, respectively (Figure 4D).»

Revision: please delete (Th) and another thing, your sentence not much the data presented in Figure 4D. You said significantly decreased for soluble sugar and soluble protein; however, I observe the reverse (or inverse), which means an increase in the soluble sugar for the mutant. Am I right!!! Please match your text to Figures 4D content.

- Line 501-502; you wrote «with four of these DEGs down-regulated»

Revision: You have action verb missing in this case. Please write as follows: with four of these DEGs were down-regulated.

- Line 682: please replace transported by transferred

- Line 715: you wrote «had been have been……»

Please correct: keep only one tense have been or had been, don’t mix things together.

I recommend authors to well check their paper before its consideration to avoid this kind of confusion.

Comments on the Quality of English Language

English is fine just few issues

Round 2

Reviewer 1 Report

Comments and Suggestions for Authors

The objective of the current study conducted by Chibuogwu et al., entitled 'Molecular Mechanisms of Chlorophyll Deficiency in Ilex × attenuata 'Sunny Foster' Mutant', was to investigate the molecular mechanisms underlying this leaf coloration. Given the clarifications provided by the authors and the clarification of the differences between the two papers, the work is now worthy of publication in its present form.